# Rapid quantitative pharmacodynamic imaging by a novel method: theory, simulation testing and proof of principle

Kevin J. Black[1], Jonathan M. Koller[2] and Brad D. Miller[2]

[1] Departments of Psychiatry, Neurology, Radiology, and Anatomy & Neurobiology, Washington University School of Medicine, St. Louis, MO, USA
[2] Department of Psychiatry, Washington University School of Medicine, St. Louis, MO, USA

## ABSTRACT

Pharmacological challenge imaging has mapped, but rarely quantified, the sensitivity of a biological system to a given drug. We describe a novel method called rapid quantitative pharmacodynamic imaging. This method combines pharmacokinetic-pharmacodynamic modeling, repeated small doses of a challenge drug over a short time scale, and functional imaging to rapidly provide quantitative estimates of drug sensitivity including $EC_{50}$ (the concentration of drug that produces half the maximum possible effect). We first test the method with simulated data, assuming a typical sigmoidal dose-response curve and assuming imperfect imaging that includes artifactual baseline signal drift and random error. With these few assumptions, rapid quantitative pharmacodynamic imaging reliably estimates $EC_{50}$ from the simulated data, except when noise overwhelms the drug effect or when the effect occurs only at high doses. In preliminary fMRI studies of primate brain using a dopamine agonist, the observed noise level is modest compared with observed drug effects, and a quantitative $EC_{50}$ can be obtained from some regional time-signal curves. Taken together, these results suggest that research and clinical applications for rapid quantitative pharmacodynamic imaging are realistic.

Corresponding author
Kevin J. Black, kevin@WUSTL.edu

## INTRODUCTION

Many important biological problems involve measuring sensitivity to a specific drug. The answers are of interest to scientists, the pharmaceutical industry, patients and clinicians. A variety of approaches have been employed. Pharmacological imaging methods, the focus of this communication, can be grouped as addressing drug sensitivity via one of two broad, nonexclusive strategies: mapping (localizing) regions of the body or of an organ that are most sensitive to drug, or measuring (quantifying) sensitivity to drug.

For many questions of interest, mapping is all that is required. One scientific example would be to identify in what part of the brain dopamine loss begins in Parkinson disease. Finding an occult cancer is a clinical example. Many methods have been employed for pharmacological mapping. Within the brain, for instance, investigators have used

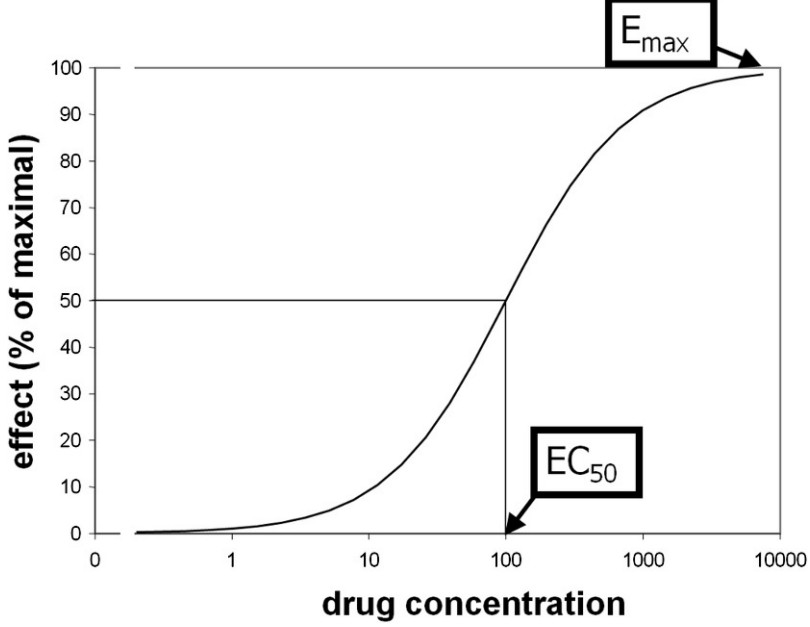

**Figure 1** Illustration of $EC_{50}$ and $E_{max}$.

autopsy studies, positron emission tomography (PET), or EEG to regionally map receptor binding or drug-induced changes in neuronal field potentials or neurotransmitter release. Pharmacological challenge functional MRI (dubbed phMRI) maps responses to a single dose of drug, usually with nonquantitative imaging methods.

Some pharmacological questions, however, do require quantification. Comparisons between groups or over time are scientific examples, and drug dose determination is a clinical and industry example. The traditional approach is to measure biological responses to different doses of drug. Standard methods to quantify receptor (or enzyme) sensitivity were derived from *ex vivo* assays such as displacing a radiolabeled ligand with varying doses of "cold" drug. Receptor binding often produces a sigmoid-shaped dose-response curve when plotted against the logarithm of drug concentration. Typically these curves reasonably fit *a priori* mathematical models and are characterized with standard parameters including $E_{max}$ (maximal magnitude of the effect of a drug at high doses) and $EC_{50}$ (plasma concentration of drug that elicits an effect half as large as $E_{max}$; see Fig. 1) (*Holford & Sheiner, 1982*). These parameters are similar to the $B_{max}$ and $K_D$ parameters from receptor binding analyses, or to models of substrate-enzyme kinetics.

The drug effect on the vertical axis of these dose-response curves can sometimes be measured clinically or by another systemic effect (e.g., change in insulin plasma concentration in response to a glucose infusion). Alternatively, one can measure responses in a single organ or even in a single cell (e.g., microdialysis in a given region of brain after administration of levodopa). Although these methods provide quantitative answers, they do not allow one to localize where the most sensitive tissues are found, at least not without numerous experiments isolating different organs or parts of an organ.

Problems with the approaches mentioned above include spatially limited information (e.g., for single-cell recordings, microdialysis, clinical or endocrine measures), information limited to one cellular level (e.g., quantification of receptors but not of second messengers), limited face validity (e.g., responses in cell culture or at autopsy) or applicability to a limited pool of subjects (e.g., PET, microdialysis, intrasurgical recordings, or autopsy).

An alternative approach addresses many of these problems. Pharmacologic activation imaging has the potential both to quantify and to map responses to drugs. Pharmacologic activation imaging refers to regional comparisons of a biological function in the presence and absence of a specific, acute pharmacological challenge. The idea is to "push on" specific receptors to see which parts of the body increase or decrease their activity, and quantify the responses. Various imaging methods have been used, and responsive regions have been mapped with increasing sensitivity (*Chen et al., 1997*; *Herscovitch, 2001*; *Perlmutter, Rowe & Lich, 1993*; *Zhang et al., 2000*).

However, to date pharmacologic activation imaging has rarely produced quantitative pharmacodynamic data such as $EC_{50}$, especially for individual subjects. This goal has been thought to require two features that are difficult or impractical with many imaging methods: truly quantitative imaging measures, and repeated imaging sessions at a number of different doses. Only rarely have data been reported that would allow an approximation of quantitative pharmacodynamic parameters (*Black et al., 2000*; *Black et al., 2002*; *Black et al., 2010*; *Hershey et al., 2000*; *Kofke et al., 2007*; *Matthews, Honey & Bullmore, 2006*).

Here we describe a novel approach that can be used to extract quantitative pharmacodynamic information from a single imaging session, even with a nonquantitative imaging modality and without any identifiable clinical effect of drug. We test the new method's performance thoroughly using simulated data, and report preliminary experiments in nonhuman primates as proof of principle.

## METHODS

### Theory

The novel approach depends on the recognition that the most accurately measured variable in most functional imaging experiments is *time*. By giving repeated doses of drug and measuring responses to each dose over time intervals short enough to minimize time-dependent artifactual signal drift, one can compute a quantitative measure of sensitivity to drug in a single imaging session, even with a nonquantitative imaging method. This process is summarized in Fig. 2. In Fig. 2A, a typical time-concentration curve for a drug is plotted using traditional pharmacokinetic modeling (black curve). To estimate the signal seen in a tissue region from a subject's imaging data, one examines the composition of the time-concentration curve with the concentration-response curve shown in Fig. 2B. (A sigmoid response is assumed here, but a different function could be chosen.) The three dose-response curves in this graph represent three different subjects (or three different tissue regions) with different sensitivity to drug. The leftmost curve, with $EC_{50} = a$, represents a region with high sensitivity (low $EC_{50}$), whereas the curves with

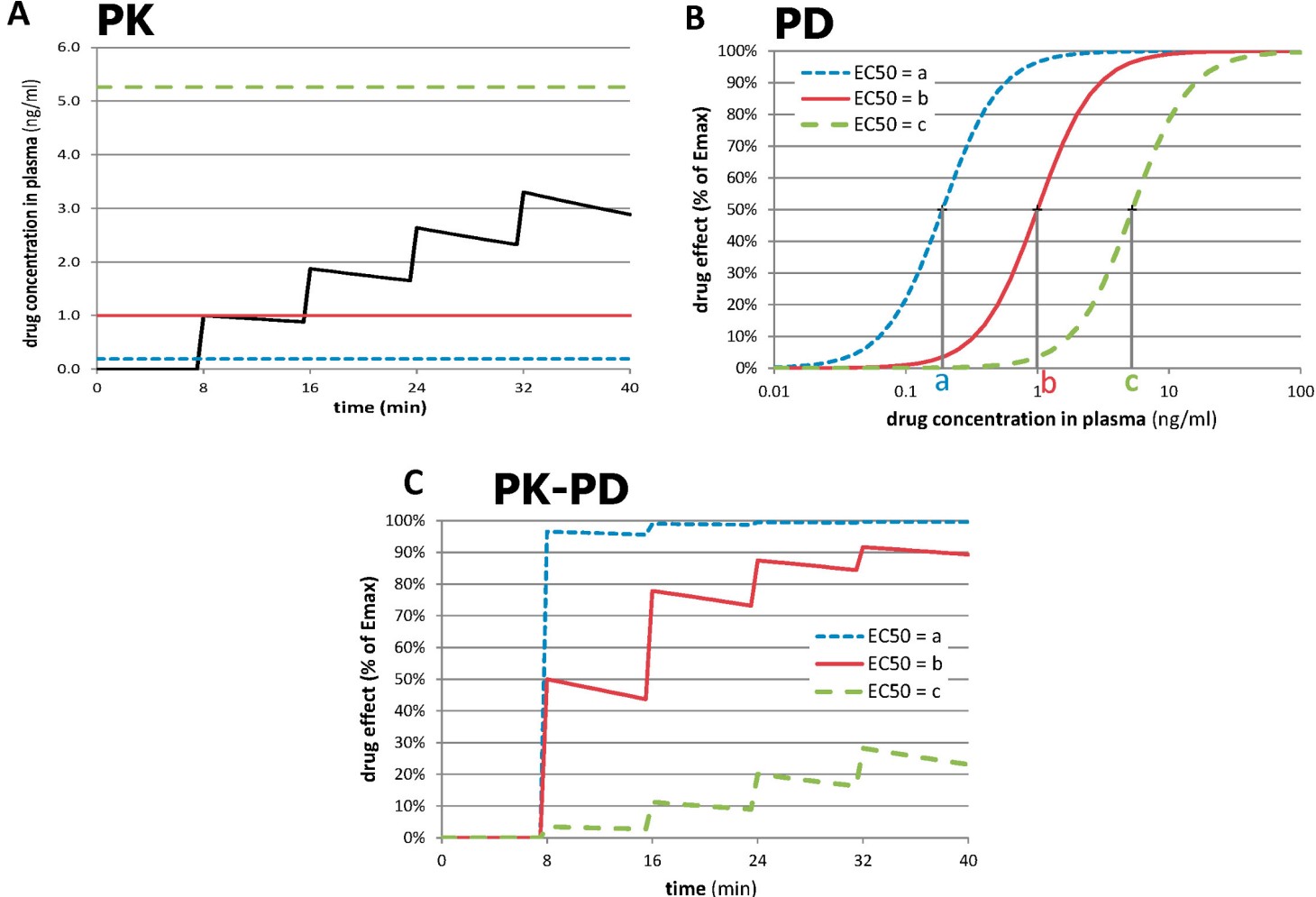

**Figure 2 Relationships between time, drug concentration, and effect in tissue for three different values of $EC_{50}$.** (A) Pharmacokinetic (PK) simulation of plasma concentration of drug over time after 4 equal doses of drug. (B) Pharmacodynamic (PD) simulation of dose-response curves from regions or subjects with different sensitivity to drug. (C) PK-PD simulation of tissue response over time. See Methods: Theory for further details. For these plots, $n = 2$, $a = 0.19$ ng/ml, $b = 1.0$ ng/ml, and $c = 5.26$ ng/ml; see Table 2 for other values.

higher $EC_{50}$s represent regions with lower sensitivity. As shown in Fig. 2C, the resulting tissue time-response curves are markedly different. For the sensitive region with $EC_{50} = a$, the first dose of drug produces a near-maximal response. For the least sensitive region, only the later doses have a noticeable effect. The following paragraphs formalize this concept.

The pharmacokinetic-pharmacodynamic (PK-PD) model describes the relationship between doses of a specific drug and the resulting effect in tissue. Initial simulation testing with this method used four intravenous drug infusions equally spaced in time during a 40-min phMRI scanning session. The pharmacokinetics are described by a simple one-compartment model with loss from plasma at a rate proportional to the plasma concentration. In the equations below, $C(t)$ describes the plasma concentration of drug over time (Table 1 summarizes the functions described in this section). $K$ doses of drug,

**Table 1  Summary of functions defined in the text.**

| Function | Description |
| --- | --- |
| $C(t)$ | Predicted plasma concentration of drug as a function of time (depends on pharmacokinetic parameters including amount and timing of drug doses, elimination half-life, and time delay $t_s$ between drug dose and plasma peak) |
| $E(C)$ | Effect of drug as a function of drug plasma concentration (depends on $EC_{50}$ and Hill coefficient $n$) |
| $E(C(t))$ | Predicted effect of drug as a function of time (composition of $E(C)$ with $C(t)$) |
| $poly_M(t)$ | A polynomial of degree $M$, $a_0 + a_1 t + a_2 t^2 + \cdots + a_M t^M$ |
| $tissue_{\text{model}}(t)$ | $= E(C(t)) + poly_M(t)$ |
| $voxel(t)$ | Sum of $tissue_{\text{model}}(t)$ and Gaussian noise (noise added independently at each time point) |

$D_k$, are given at times $t_k$, and $u(t)$ is the unit step function. The pharmacokinetic model also includes a fixed time delay (shift) $t_s$ and a half-life $t_{1/2}$ for loss of drug from plasma.

$$C(t) = \sum_{k=1}^{K} D_k \cdot 0.5^{\left(\frac{t - t_s - t_k}{t_{1/2}}\right)} \cdot u(t - t_s - t_k).$$

This plasma concentration curve $C(t)$ then becomes the input to a traditional sigmoid concentration-effect model to describe the pharmacodynamics (*Holford & Sheiner, 1982*):

$$E(C) = \frac{E_{\text{max}} C^n}{EC_{50}^n + C^n}.$$

This model is characterized by the maximal effect of drug at high doses, $E_{\text{max}}$; the Hill coefficient $n$, which models the number of drug molecules required to activate the receptor and can be understood as describing the steepness of the sigmoid curve at its inflection point; and $EC_{50}$, which is the drug concentration that produces an effect half as large as $E_{\text{max}}$. Figure 1 shows this curve on a logarithmic $x$ axis. The input to the curve is plasma concentration of a drug, and the effect that is measured and modeled is the imaging signal. Figure 3 shows several predicted tissue time-activity curves $E(C(t))$ in response to drug administration, based on different values of the PK-PD parameters.

Table 2 lists the model parameters used to generate the simulated data. To account for nonquantitative signal drift encountered with BOLD-sensitive fMRI, unrelated to drug administration, the model then adds signal drift modeled by a polynomial $poly_M(t)$ of degree $M$. The sum $E(C(t)) + poly_M(t)$ comprises the model for the imaging signal, here called $tissue_{\text{model}}(t)$.

## Simulation: generation of test data

Predicted time–imaging signal curves $tissue_{\text{model}}(t)$ were generated for a wide range of plausible values for $EC_{50}$, half-life, and other PK-PD model parameters. Then Gaussian noise was added to $tissue_{\text{model}}(t)$ to better reflect real-world situations. A random noise function of time was generated 1000 times for each level of noise to allow testing of sensitivity and specificity of the curve-fitting methods. The level of noise was quantified by the standard deviation (SD) of the noise across a given time series, and is reported

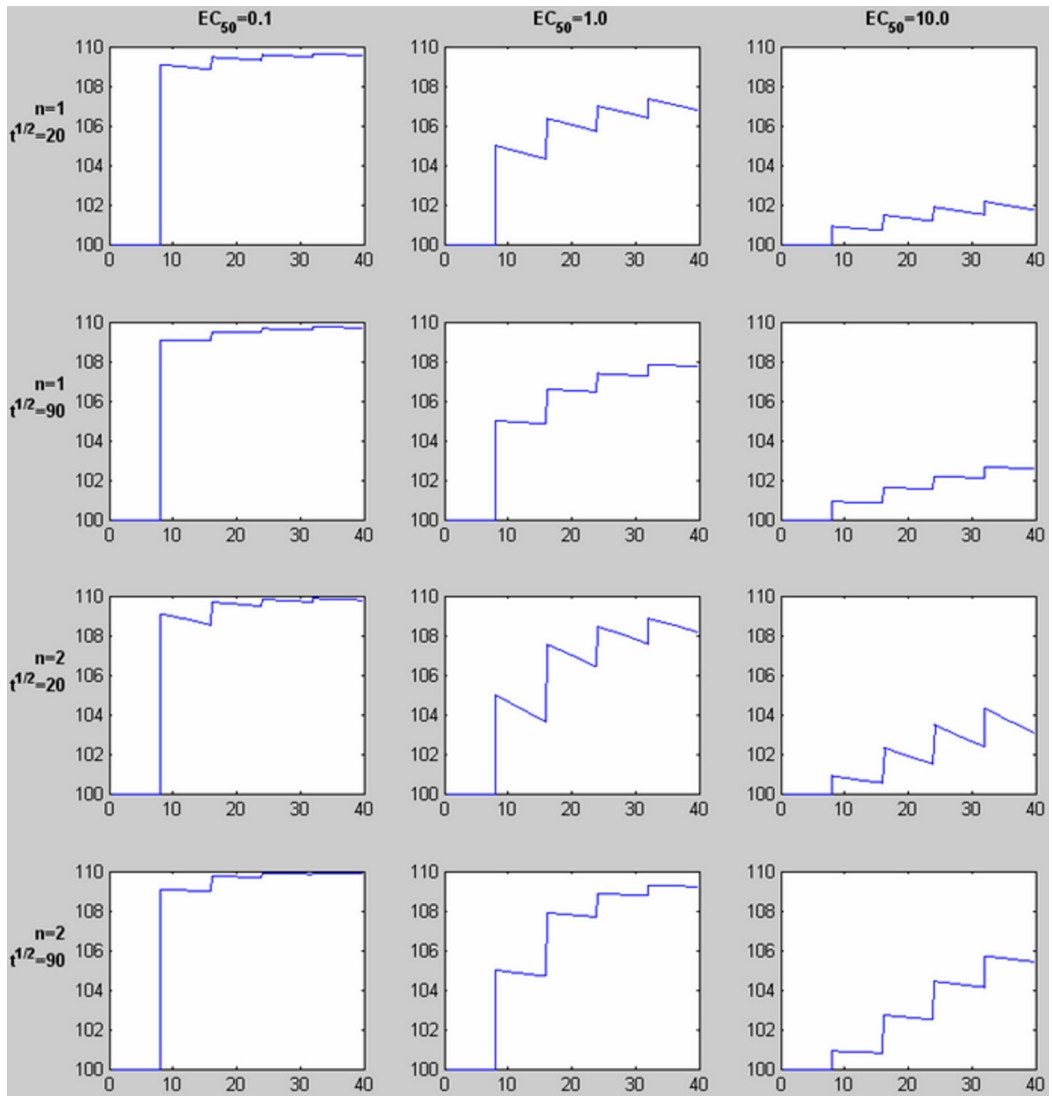

**Figure 3 Time-signal curves predicted by different values of $EC_{50}$, the Hill coefficient $n$, and drug elimination half-life $t_{1/2}$.** For each graph, 4 equal doses of drug are assumed, with the first dose producing a plasma concentration of 1.0 (given in the same units as $EC_{50}$, e.g., ng/ml). The horizontal axis shows time in the same units as $t_{1/2}$ (e.g., minutes). The vertical axis shows a response with $E_{max} = 10$ added to a constant baseline signal of 100. Note the shape of the time-signal curve, not just the amplitude, varies substantially depending on the values of $EC_{50}$, $n$, and $t_{1/2}$.

relative to the known (input) $E_{max}$. Below we refer to the sum of noise and $tissue_{model}(t)$ as $voxel(t)$.

Figure 4 shows examples of $tissue_{model}(t)$ (solid line) and $voxel(t)$ (dots) for different levels of noise. At high levels of noise, curve fitting will obviously be difficult.

To test how likely the curve-fitting procedure was to report a fit to data in the absence of an (intentional) signal, noise was also repeatedly generated and added to a polynomial

**Table 2 Values used for each parameter in the model.** The values shown here were used to generate the $E(C(t))$ curves used for the final simulation testing.

| Name | Description | Values used in the simulated data | Units |
|---|---|---|---|
| $K$ | Number of drug doses | 4 | (None) |
| $D_k$ | Magnitudes of each drug dose | Not simulated; equal doses, with the first dose producing a plasma concentration peak $= 1$ | arbitrary (e.g., ng) |
| $t$ | Time | from 0 to 40 | min |
| $t_k$ | Time of drug dose $k$ | 8, 16, 24, and 32 | min |
| $t_s$ | Fixed time delay (shift) | 0.43 | min |
| $t_{1/2}$ | Elimination half-life for loss of drug from plasma | 41 | min |
| $E_{max}$ | Maximal possible (asymptotic) effect of drug | 10 | Arbitrary imaging units (e.g., scaled MRI signal intensity) |
| $EC_{50}$ | Concentration producing 50% of the maximal possible effect, $E_{max}/2$ (when Hill coefficient $n = 1$) | One of the following values for each data set: 0.1, 0.43, 1.0, 1.7, 3.0, 3.8, 5.0, 6.1, 8.0, 9.2 | Arbitrary plasma concentration units (e.g., ng/mL) |
| $n$ | Hill coefficient | 1 | (none) |
| $M$ | Degree of noise polynomial $poly_M(t)$ | 1 | (none) |
| $a_k$ | Coefficients of $poly_M(t)$ | 1000, 0.05; i.e., $y = 1000 + 0.05t$ | $a_0$: arbitrary imaging units (e.g., scaled MRI signal intensity) $a_1$: (imaging units) $\cdot$ min$^{-1}$ $a_k$: (imaging units) $\cdot$ min$^{-k}$ |

of low degree, resulting in $voxel(t)$ curves in which no drug response was included (i.e., $E_{max} = 0$). Without noise the algorithm cannot fit another curve better than the original polynomial. With substantial noise, however, it is possible that the resulting data may be fit better by a curve that includes modeled drug response (a false positive fit to the data).

These simulated data sets (1000 instances of $voxel(t)$ for each set of parameter values and noise level) are available at datadryad.org (*Black, Koller & Miller, 2013*).

### Parameter estimation

A custom computer program (QuanDyn$^{TM}$, available from the authors) was written for Windows XP in Microsoft Visual Basic. It estimates which parameters for the above PK/PD model best fit an input time series. The program repeats this process at each voxel of a 4D imaging dataset. The program first temporally filters the data, replacing the data at each time point with the median of the data acquired over a longer interval centered on that time point. For the present analysis a 45-s interval was used.

To speed and simplify the simulation testing described below, and because our original curve-fitting algorithms performed poorly for some parameters, we allowed each PK/PD model parameter approximately 10 values. In other words, for the purposes of this testing, the program could produce only certain answers (Table 3). These values were appropriate to the range of input data used in simulation testing. This quantization was felt to be reasonable since in practice precision is adequate if the $EC_{50}$ reasonably approximates the achieved blood concentration. For testing, in half the cases the input ("correct") answer was chosen to be a value not available to the program, so as not to favorably bias the results.

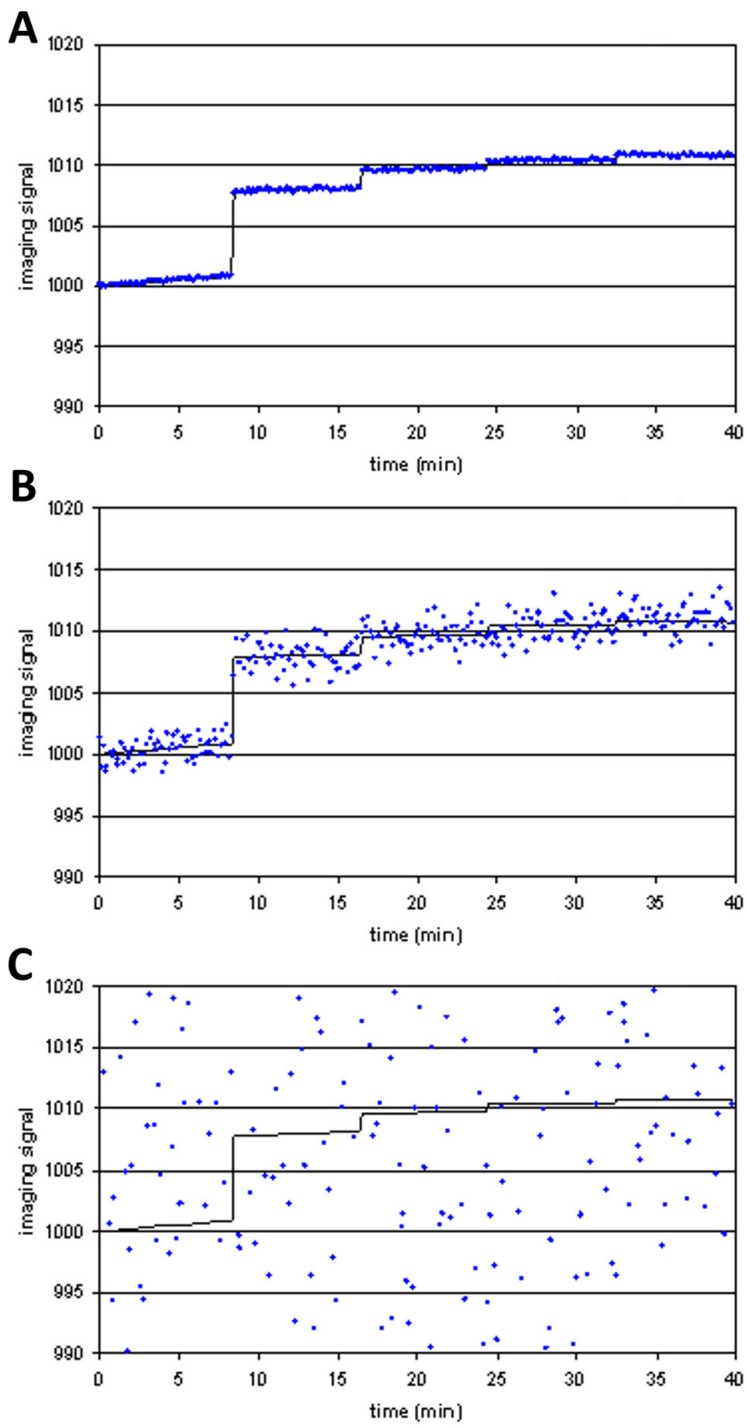

**Figure 4 Three noise levels added to the same simulated imaging signal.** (A) $SD = 0.01 \cdot E_{max}$; (B) $SD = 0.1 \cdot E_{max}$; (C) $SD = 2 \cdot E_{max}$.

**Table 3 Output range for each parameter fit by the model when analyzing the simulated data.** For each parameter, the simulation testing produced a value selected from the output range shown in the table. In this table, $[a, b]$ indicates a closed interval on the real line, $\mathbb{R}$ indicates the set of all real numbers, and $\{a, b, c, \ldots\}$ indicates a list of allowed values.

| Name | Description | Starting value | Output range | Units |
|------|-------------|----------------|--------------|-------|
| $t_s$ | Fixed time delay (shift) | n/a[*] | $\{0, 0.1, 0.2, 0.3, 0.4, 0.5, 0.6, 0.7, 0.8, 0.9, 1.0\}$ | min |
| $t_{1/2}$ | Elimination half-life for loss of drug from plasma | 41 | $\{41\}$ | min |
| $E_{max}$ | Maximal possible (asymptotic) effect of drug | n/a[**] | $\mathbb{R}$ | Arbitrary imaging units (e.g., scaled MRI signal intensity) |
| $EC_{50}$ | Plasma concentration producing half-maximal effect $E_{max}/2$ (when Hill coefficient $n = 1$) | n/a[*] | $\{0.1, 0.5, 1, 2, 3, 4, 5, 6.5, 8, 10\}$ | Arbitrary plasma concentration units (e.g., ng/mL) |
| $N$ | Hill coefficient | 1 | $\{1\}$ | None |
| $M$ | Degree of noise polynomial $poly_M(t)$ | 2 | $\{2\}$ | None |
| $a_i$ | Coefficients of $poly_M(t)$ | n/a[**] | $\mathbb{R}$ | None |

**Notes.**

[*] Tested at each of the values listed.

[**] Computed directly by least squares fit for each set of other parameters evaluated.

The cost function minimized by the program was the summed squared error of the model compared to the time-signal curve $voxel(t)$.

The parameter $EC_{50}$ refers to a concentration of drug in plasma. For this simulation, the peak plasma concentration attained after the first bolus of drug was taken as 1 unit, and $EC_{50}$ was computed relative to that concentration. In a biological system, the computed relative $EC_{50}$ values could be easily scaled to absolute $EC_{50}$s by multiplying by the actual plasma concentration of drug sampled at the appropriate time.

After experimentation with iterative, linear and nonlinear optimization of the parameters of interest, a combination of iterative and linear (least squares) curve-fitting was found to supply a reasonably accurate yet efficient numerical methods solution (see Results: model fitting).

## Statistical test of goodness of model fit to test data

Summed squared error across all time points at a given voxel was used to quantify how well the data at that voxel, $voxel(t)$, were fit by a time-activity curve $tissue_{model}(t)$ generated by the PK-PD model using a given set of model parameters. A comparison function that did not incorporate any information about the timing of drug administration was used to statistically test how well the model fit the data. For simplicity the comparison function chosen was a polynomial with the same number of degrees of freedom as the PK-PD model $tissue_{model}(t)$. Specifically, if $j$ PK-PD model parameters were used to generate $E(C(t))$, to which a polynomial of degree $M$, $poly_M(t)$, was added to give $tissue_{model}(t)$, then the comparison function was $poly_N(t)$, where $N = M + j$. The ratio $F$ of the summed squared error for $poly_N(t)$ to the summed squared error for $tissue_{model}(t)$ is computed and saved to create a statistical image reflecting the improvement in the fit to the data by incorporating knowledge of drug administration times and the PK/PD model. Higher

values for $F$ indicate better fit for the model. The probability that the model fit better than the comparison polynomial by chance was computed by interpreting $F$ as an $F$ test statistic with $j$ and $([\text{number of time points in } voxel(t)] - j - 2)$ degrees of freedom.

## Simulation testing: test statistics

For each combination of parameters tested, the model was fit to 1000 independently generated $voxel(t)$ time-signal curves at each noise level. Since only certain values were possible for $EC_{50}$, continuous statistics like mean $\pm$ SD are not appropriate. The following summary statistics were used to characterize each pair of $EC_{50}$ and noise values:

- **model fit sensitivity** = the frequency with which $F$ exceeded a set threshold in 1000 $voxel(t)$ curves generated with the same set of parameters and noise SD
- **model fit specificity** = one minus the frequency with which $F$ exceeded the threshold in the (polynomial + noise) data.
- **$EC_{50}$ sensitivity** = fraction of times (of 1000) that the primary parameter of interest ($EC_{50}$) returned by the program was "correct", i.e., if the value of $EC_{50}$ used to generate the data was 0.43, and the possible answers included $\{\ldots, 0.1, 0.5, 1, \ldots\}$, then only the nearest possible answers, 0.1 and 0.5, were counted as correct. In other words, how often does the program return (as close as possible to) the desired value?
- **$EC_{50}$ specificity** was computed from the (polynomial + noise) data, and was taken to be $1 - p$, where $p$ is the fraction of times that the program returned a given value for $EC_{50}$.
- **$EC_{50}$ positive predictive value (PPV)** was defined as follows. A given allowed output value of $EC_{50}$ could have been returned by the program from input generated from a different $EC_{50}$ value plus noise. The PPV was computed as a fraction whose denominator was equal to the number of times that the given $EC_{50}$ value was output by the program across the tens of thousands of voxels generated by all sets of input model parameters, and whose numerator was equal to the number of those times when the output was the correct answer for the input. For example, of all the times the program output $EC_{50} = 0.5$, for what fraction was that the correct answer (i.e., what fraction came from an input $EC_{50}$ value nearer 0.5 than any other allowed output value)?

It should be noted that PPV depends on the prior probability, i.e., how often the specified output value was supposed to be produced, so the actual PPV in a given experimental situation may differ from that computed here. However, the PPV can be computed from prior probability, sensitivity and specificity when these are known.

## Biological data: protocol

These studies were approved by the Washington University Animal Studies Committee (protocols # 20020085, 20050126). Two male macaques (*M. fascicularis*) were studied under inhaled anesthesia (1.0–2.0% isoflurane, titrated individually by a veterinary technician to the minimal level consistent with continued sedation). A 20 g plastic catheter was inserted over a needle into a lower extremity vein. Just prior to drug infusion, the catheter and tubing's known volume was filled with drug solution. Repeated

BOLD-sensitive asymmetric spin echo fMRI images were obtained on a Siemens 3.0T Allegra magnet with a custom head coil. Over a 40-min period, 800 whole-brain image volumes were obtained, one every three seconds. A 3D T1-weighted structural image was also obtained for anatomical comparison (*Mugler & Brookeman, 1990*). Images were transformed into the macaque atlas space of Martin and Bowden (*Martin & Bowden, 1996*; *Martin & Bowden, 2000*; www.purl.org/net/kbmd/cyno) using previously validated methods (*Black et al., 2004*).

## Biological data: single-dose experiment

In one experiment (single-dose experiment), at 15 min into the BOLD imaging, an intravenous infusion of the dopamine D1 agonist SKF82958 was begun, and 0.1 mg/kg was infused at a fixed rate over 5 min. We have previously shown that this dose of drug does not alter whole-brain average quantitative blood flow (*Black et al., 2000*), ruling out a meaningful direct vascular effect of this drug and implying that regional changes in hemodynamic signal most likely reflect true changes in regional metabolic activity.

Midbrain and (average left and right) striatal volumes of interest (VOIs) were drawn on the anatomical image. The midbrain is the region of brain with the highest regional sensitivity to exogenous levodopa (*Black et al., 2005*; *Hershey et al., 2003*; *Trugman & Wooten, 1986*), and striatum has been examined in several prior phMRI studies. The midbrain region measured 0.17 mL and the striatal region 0.80 mL. For each VOI, the average time-signal curve was extracted and temporally smoothed with a 45-s median filter to match the QuanDyn$^{TM}$ filtering. Additionally, a noise/effect ratio was computed to allow comparison to noise/$E_{max}$ from the simulated data. To this end, a regression line was fit to the first 15 min of data, during which no drug was administered, and the standard deviation of the residual signal was computed and divided by the maximal signal produced by the drug infusion. By definition, the drug-induced signal seen with this dose cannot exceed the maximal possible signal $E_{max}$, implying that the desired ratio, (SD of residuals)/$E_{max}$, is probably smaller and cannot be worse than the estimate (SD of residuals)/(peak signal change after drug) derived from this experiment.

## Biological data: multiple-dose proof-of-principle experiments

In other experiments (multiple-dose proof-of-principle experiments), the same total dose of SKF82958 was separated into 4 or 8 equal aliquots, each of which was infused over 30 s at 4-min intervals. Each animal had a 4-dose and an 8-dose study. The QuanDyn$^{TM}$ program was applied to these data using both a 30-min half-life (*Black et al., 2000*) and a 5-min half-life, in case there were prominent distribution effects. The $t_{1/2}$ value that produced the higher F statistic was retained.

# RESULTS

## Simulation: model fitting

### Shape of cost function in parameter space

Plots of the cost function surface over bivariate plots of $EC_{50}$ and time shift, without noise, showed well-behaved error with a single clear minimum at the correct value. Similar plots

with $EC_{50}$ and either half-life or the Hill coefficient $n$ showed flatter surfaces, suggesting half-life or $n$ would be harder to fit even with perfect data. Half-life can be determined without imaging methods and $n$ can be reasonably estimated from *in vitro* data. The results below were derived assuming a fixed half-life and $n = 1$.

### Model fit sensitivity

At low levels of noise, e.g., SD $< 0.05E_{\max}$, the PK-PD model always fit the data better than the null hypothesis (polynomial) regardless of the input $EC_{50}$, i.e., $F$ exceeded 1.218, the value at which $p(F) = 0.05$, in all or nearly all voxels tested. See Fig. 5A. At high levels of noise, e.g., SD $> 0.5E_{\max}$, the model rarely fit significantly better than the null hypothesis. (For comparison, different ratios of SD to $E_{\max}$ are shown in Fig. 4.) At realistic intermediate values of noise, the model fit better for lower $EC_{50}$ values (i.e., more sensitive regions/subjects). Similar results obtained when $F$ was thresholded at the Bonferroni corrected value for 64,000 voxels (typical for a brain image); see Fig. 5B.

### Model fit specificity

Here the question is how often the program identifies a voxel as fitting the PK-PD model when the voxel contains no drug signal but only noise. Formally, the question is how often the model fits *voxel*($t$) significantly better than the null hypothesis polynomial does (judging by the $F$ statistic, at the $p = 0.05$ level, uncorrected for multiple comparisons). This did not happen once among all the thousands of time-activity curves generated, giving a model-fitting specificity of 100%. With a sensitivity of 100%, this implies a model fit PPV of 100% with these test data. In other words, voxels whose F statistic exceeds the value corresponding to $p = 0.05$ are very unlikely to come from noise of the type modeled here.

In summary, the model fits the data well under these conditions. The next question is whether the answer is correct, i.e., whether the $EC_{50}$ that produces the best-fitting model curve is accurate.

## Simulation: accuracy

### $EC_{50}$ sensitivity

Figure 6 shows how often the calculated $EC_{50}$ is correct as a function of input $EC_{50}$ and noise level. Since for this analysis we limited the QuanDyn$^{\text{TM}}$ software to producing one of 10 possible values, the question is framed as follows. If the input $EC_{50}$ fell in the interval between two possible output values, does the program return one of those two values? As seen in Fig. 6A, with small amounts of noise the software nearly always returns the correct $EC_{50}$. As noise increases, the model is less likely to find the correct $EC_{50}$, especially for high $EC_{50}$s (i.e., regions relatively insensitive to drug), where the signal is fainter. Limiting the analysis to voxels identified as fitting the model significantly better than a null-hypothesis curve improves the results somewhat (Fig. 6B).

### $EC_{50}$ specificity

Given a specified level of noise added to a null hypothesis curve (a polynomial), how likely is it that the quantitative pharmacodynamic method will return a given value for $EC_{50}$?

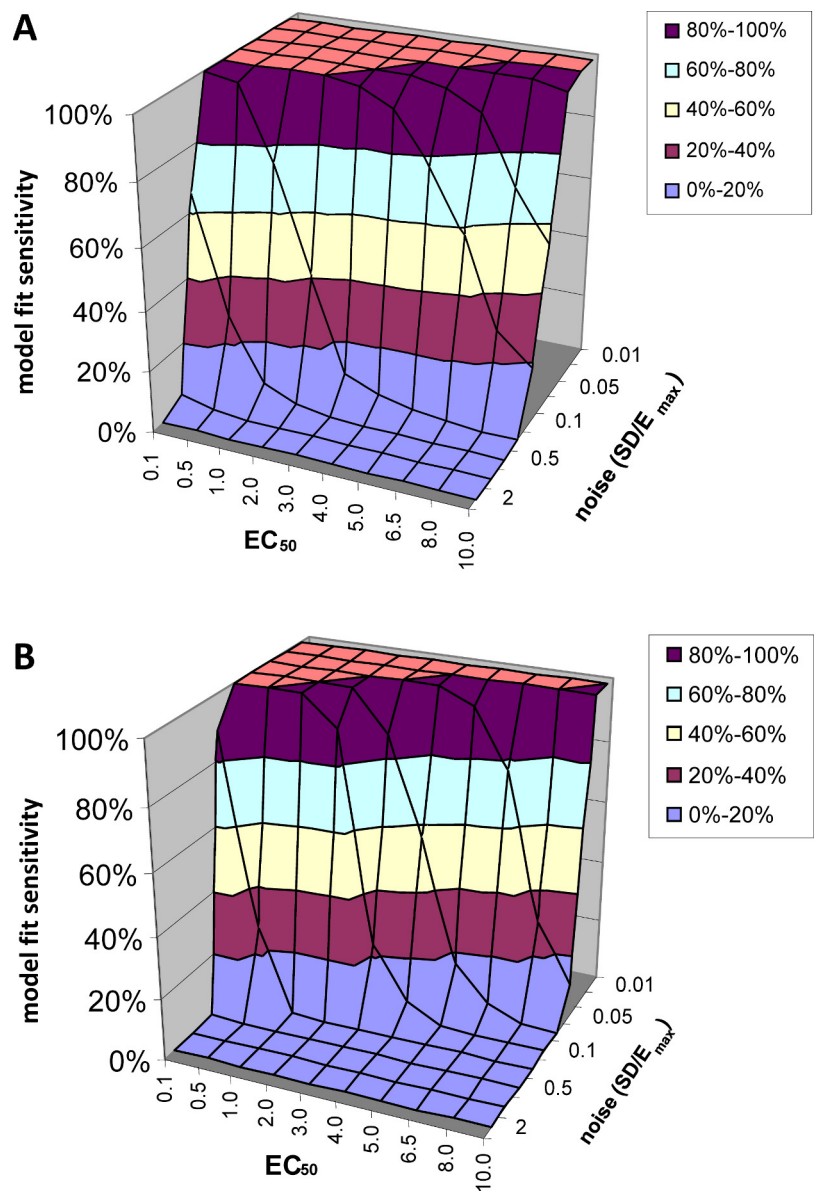

**Figure 5 Model fit sensitivity as a function of $EC_{50}$ and noise.** Vertical axis shows fraction of input curves (with noise) for which $F$ exceeds the $F$ statistic value corresponding to $p = 0.05$, as a function of input $EC_{50}$ and noise level. In (A), the $F$ threshold corresponds to $p = 0.05$ uncorrected. In (B), the $F$ threshold corresponds to $p = 0.05/64000$, representing Bonferroni correction for an image of 64,000 voxels, such as an MR image of human brain.

Surprisingly, this was relatively insensitive to the amount of noise added to the polynomial. About 30% of the time the program returned the lowest allowed $EC_{50}$, about 42% of the time it returned the highest allowed $EC_{50}$, and the remaining results were fairly evenly scattered among the other possible output values for $EC_{50}$. Fortunately, as noted above, in these cases the model never fit the data better than a polynomial, so specificity was 100% after censoring results with $F$ below threshold.

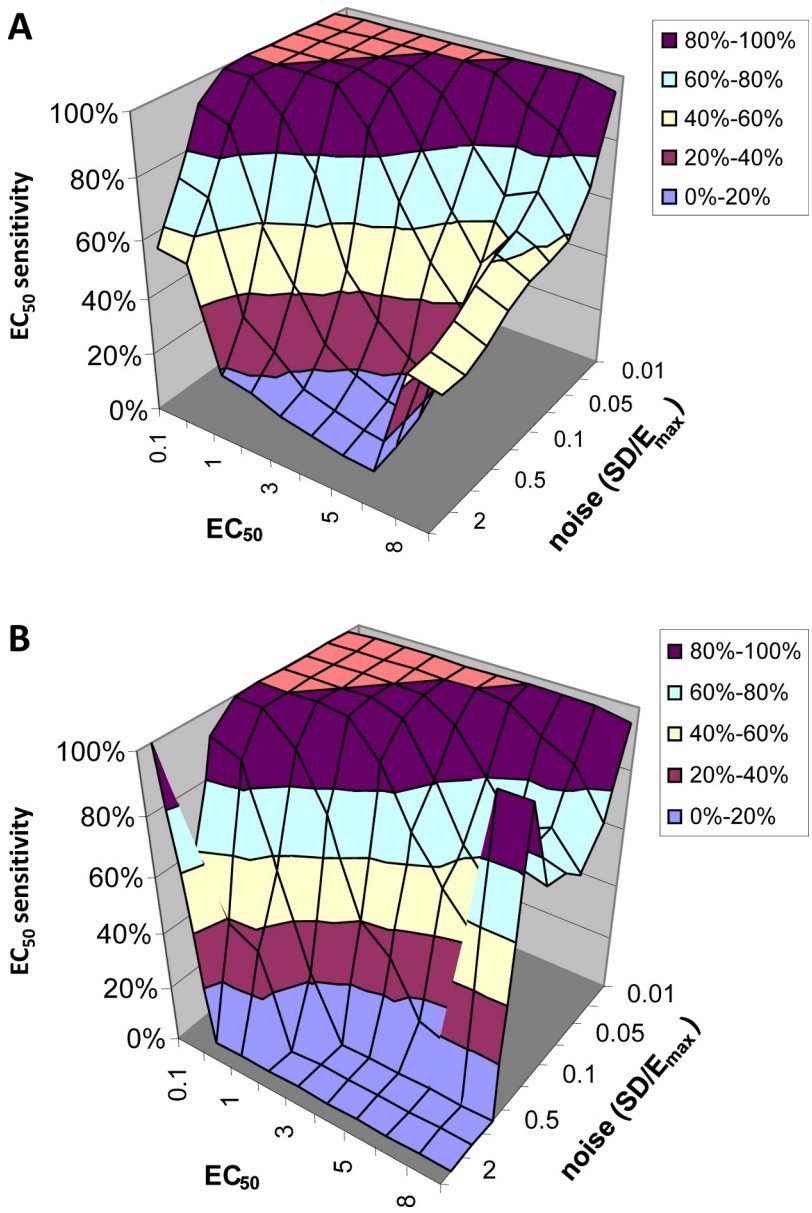

**Figure 6** *$EC_{50}$ sensitivity as a function of $EC_{50}$ and noise.* Vertical axis shows percentage of voxels whose output $EC_{50}$ was within range of the input $EC_{50}$. (A) includes all voxels. (B) includes only those voxels in which the model fit the data significantly ($F > 1.218$).

### $EC_{50}$ positive predictive value

The next result is a measure of how confident one can be in the estimate of $EC_{50}$ returned by this method. Across all 90,000 voxels generated from all selected values of $EC_{50}$ and other parameters and all levels of noise, we computed the answer to the following question. If the QuanDyn$^{TM}$ software returns a given value for $EC_{50}$, what is the likelihood that value is "in range", i.e., that it was computed from data generated with an $EC_{50}$ between the next lowest and next highest possible output values? The results are shown in Fig. 7.

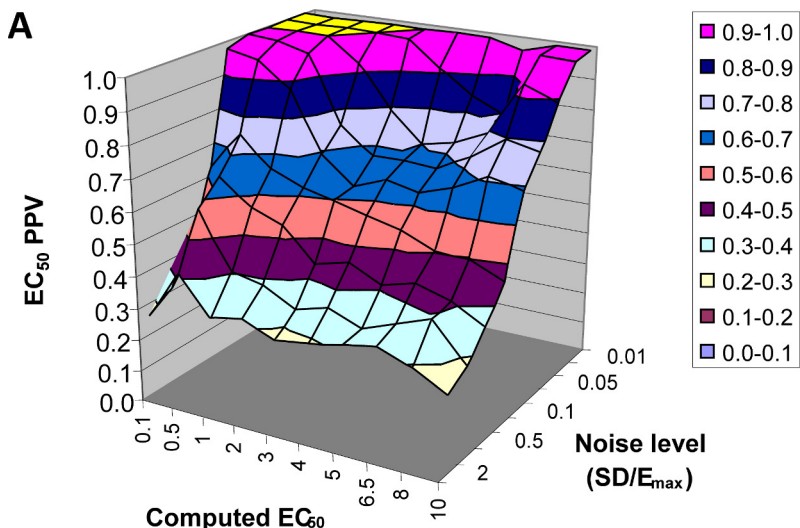

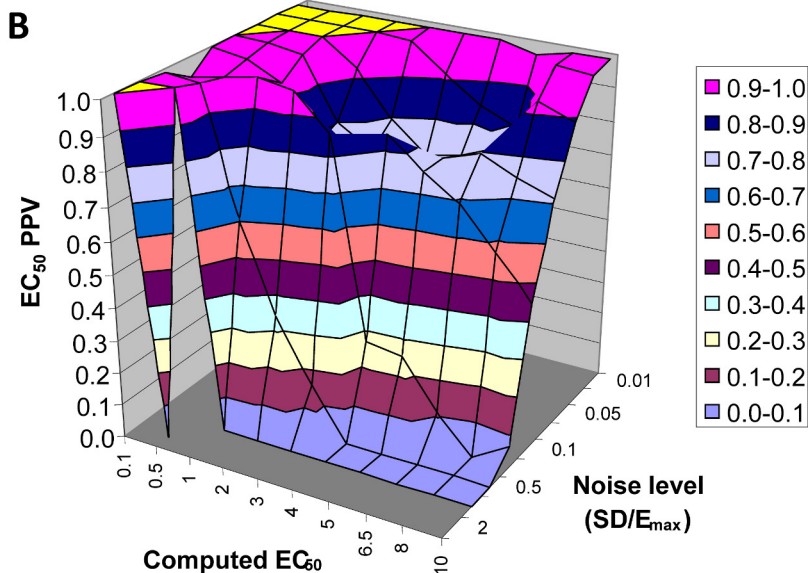

**Figure 7** $EC_{50}$ **positive predictive value.** The PPV is a measure of how confident one can be in the computed $EC_{50}$ result. (A) Data from all voxels tested. (B) Results from only those voxels in which the model fit the data significantly ($F > 1.218$).

In summary, this method usually gives a good answer for $EC_{50}$ in the simulated data as long as noise is low or $EC_{50}$ is low, indicating reasonable sensitivity to drug. The natural next question is, can we expect noise this low in biological data?

## Signal vs noise estimation, with biological data (single-dose experiment)

An intravenous infusion of a dopamine D1 receptor agonist in a dose that does not affect whole-brain mean blood flow induced a clear signal (about 2.2% of modal brain BOLD

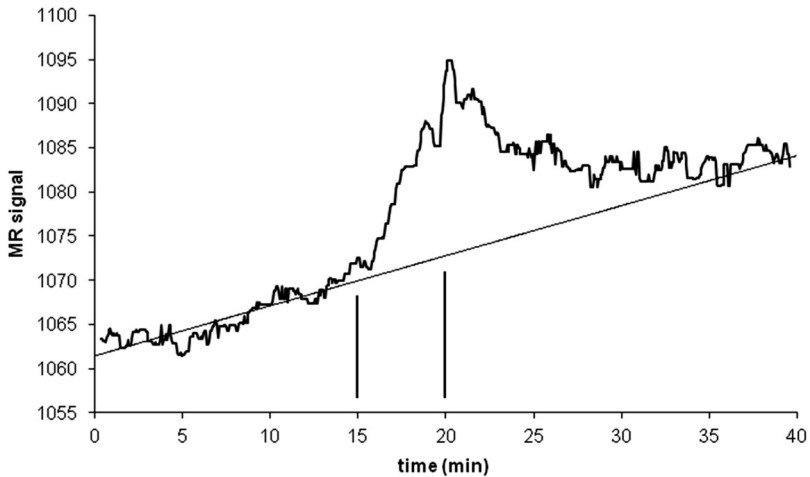

**Figure 8 Time-signal curve in an *a priori* midbrain VOI from the single-dose fMRI experiment described in Methods.** A single injection of dopamine $D_1$ receptor agonist was given slow i.v. from time 15 to 20 min (vertical lines). The regression line shown was fit only to the baseline data (time 0–15 min).

signal) in an *a priori* VOI (Fig. 8). The drug-induced signal showed a pharmacokinetically reasonable time course, peaking at the end of the infusion, when plasma concentration of drug is highest, and largely fading over the next 15 min. The SD of the residual signal after fitting a line to the VOI data from the baseline period preceding the drug infusion was clearly much smaller than the drug-induced signal change. Since by definition $E_{max}$ is at least as large as the signal observed at any given dose, we can compute for each region an upper bound, SD/(observed effect), on the ratio SD/$E_{max}$. For the midbrain VOI the ratio was 0.05 in one animal (Fig. 8) and 0.08 in the other. The striatal VOI gave a ratio of 0.06 in the second animal, but in the first animal there was negligible striatal response to the drug, giving a ratio of 0.93. Fig. 9 shows that in simulated data, noise-to-$E_{max}$ ratios of 0.05–0.08 predict a PPV of 70–100% depending upon $EC_{50}$. Even these are underestimates for PPV, since the observed effect is a lower bound for the maximum possible effect $E_{max}$.

## Estimation of $EC_{50}$ in primate (multiple-dose proof-of-principle experiments)

We applied the method to 8 regional time-signal curves: a midbrain and a striatum VOI in a 4- and an 8-dose experiment in each of two animals. For 6 of the 8 time-signal curves, the F statistic was less than 1.2, indicating that the model did not fit the data better than chance and the $EC_{50}$ estimates should be rejected (see Table 4 and Figs. 10A, 10B).

One animal had two regions with $F > 1.218$, namely the midbrain in the 4-dose experiment and the striatum in the 8-dose experiment (see Table 4 and Figs. 10C, 10D). The two $EC_{50}$ estimates are approximately 8 and 5 times the peak blood level after the first 25 μg/kg i.v. dose. If we had a quantitative measurement of that blood level, e.g., in ng/mL, then the $EC_{50}$ values would also be absolute estimates in ng/mL.

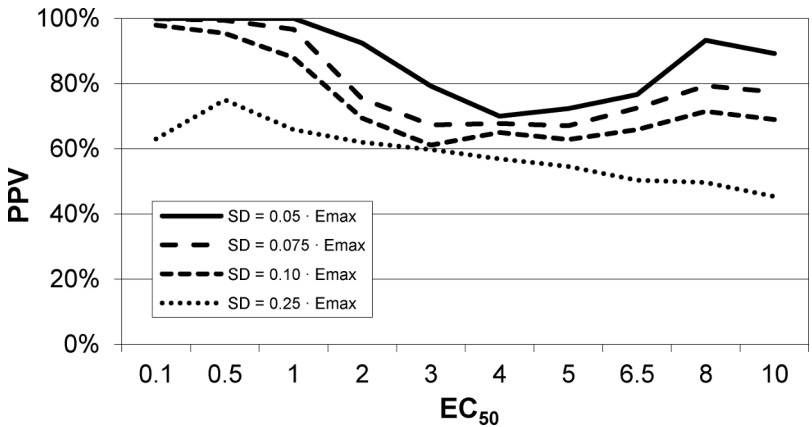

**Figure 9** $EC_{50}$ **positive predictive value for four noise levels, based on simulated data.** By comparison, the noise level in the fMRI experiment shown in Fig. 8 was $\leq 0.06 \cdot E_{\max}$, nearest the uppermost curve shown here.

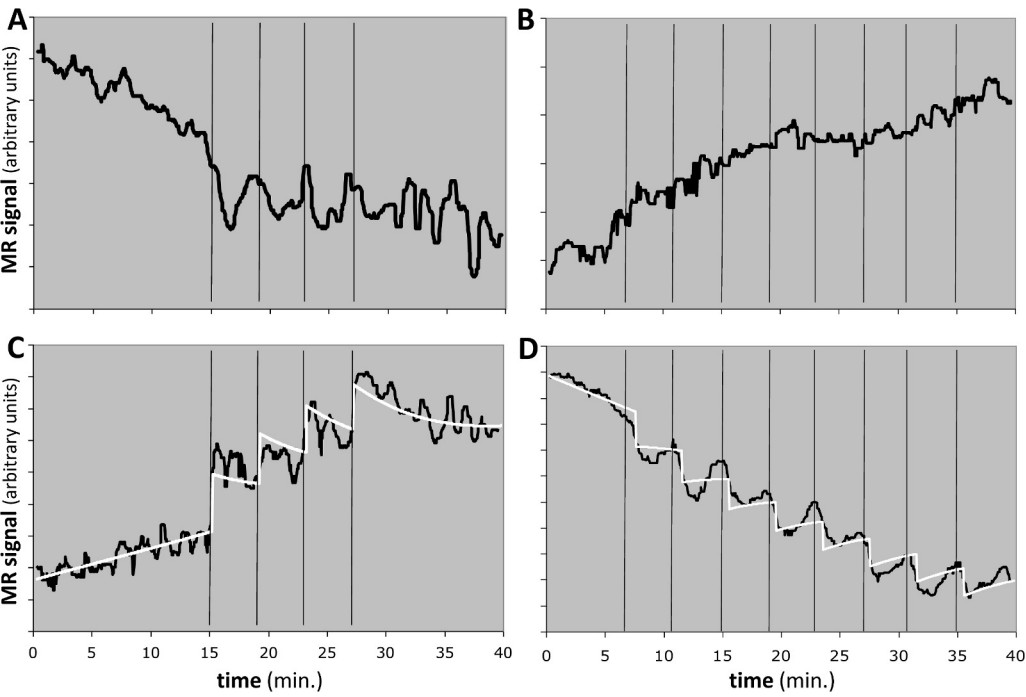

**Figure 10 Response to the dopamine $D_1$ agonist SKF82958 in four experiments from two animals.** Horizontal axis shows time in minutes. Vertical black bars indicate successive fractional doses of drug. Black curve indicates observed fMRI signal. White curve in (C) and (D) shows the best model fit to the data. Total dose in each case was 100 µg/kg i.v. See Table 4 for further details. (A, B) Time-signal curves for two experiments in which the model did not fit the data significantly better than did a polynomial with the same degrees of freedom. (C) Computed $EC_{50} = 8 \cdot X$, where $X$ = peak blood level after 25 µg/kg i.v. (D) Computed $EC_{50} = 10 \cdot Y \approx 5 \cdot X$, where $Y$ = peak blood level after 12.5 µg/kg i.v.

**Table 4 Results from the multiple-dose proof-of-principle experiments.**

| Doses | Animal | Region | F | Comments |
|---|---|---|---|---|
| 4 | 1 | Midbrain | 0.813 | – |
| 4 | 1 | Striatum | 1.132 | – |
| 4 | 2 | Midbrain | 1.884* | $t_{1/2} = 5$ min, $EC_{50} = 8$ times the peak blood level after 25 µg/kg, $E_{max} = 32.3$; Fig. 10C |
| 4 | 2 | Striatum | 0.992 | Fig. 10A |
| 8 | 1 | Midbrain | 0.973 | – |
| 8 | 1 | Striatum | 0.991 | – |
| 8 | 2 | Midbrain | 0.724 | Fig. 10B |
| 8 | 2 | Striatum | 1.570* | $t_{1/2} = 5$ min, $EC_{50} = 10$ times the peak blood level after 12.5 µg/kg, $E_{max} = -29.6$; Fig. 10D |

Notes.

* $p < 0.05$ for model fit to data.

## DISCUSSION

In simulated data, this novel quantitative pharmacodynamics method performed well. Under a reasonable set of assumptions, this method returns the correct answer either if it claims a given region has high sensitivity to drug effect (i.e., low $EC_{50}$), or if noise is of modest magnitude relative to the drug-induced signal. The assumptions used for this simulation are reasonable: (1) the object imaged has a response to the drug that can be detected by the imaging method employed; (2) the sigmoid response model is appropriate for the drug effect being studied; (3) random error is reasonably approximated by a normal distribution; and (4) nonrandom error (signal drift) can be reasonably modeled by a low-degree polynomial. In this scenario, not every possible model parameter can be simultaneously fit to the data accurately, but one can reasonably assume a single value for $n$, and half-life can be measured directly.

These simulation results suggested that the key remaining question was the actual relative magnitudes of imaging system noise and a realistic drug-induced physiological signal. Our fMRI data in a nonhuman primate give a real example of a system in which a drug-induced imaging signal is large with respect to baseline fluctuations in the same volume of interest (Fig. 8). With a signal:noise ratio of this magnitude, simulations predict a high degree of confidence in $EC_{50}$ estimates from this method (Fig. 9).

In an initial proof-of-principle study using a dopamine D1 agonist in nonhuman primates, the model fit the data significantly in two regional time-signal curves and the method provided a quantitative estimate for $EC_{50}$ for each region (relative to the peak concentration of drug in blood after the first dose). Remarkably, the estimated $EC_{50}$s were *higher* than the peak blood level of drug in these experiments, or, put another way, this method could estimate the $EC_{50}$ without needing to give high doses of drug that produce anywhere near a maximal effect (not to mention a higher chance of noxious side effects).

This new method provides for the first time a quantitative pharmacodynamic measure from a single imaging session, even with an imaging method subject to baseline signal drift.

This advance can potentially allow research and clinical applications that are not possible from qualitative methods.

## Comparison to prior methods

Though other approaches have been suggested (*Schwarz et al., 2007*), most prior research with nonquantitative imaging methods has used two general strategies to map responses to a single dose of drug. One mapping strategy could be called the pure pharmacokinetic approach, which identifies voxels whose time-signal curve approximates that of the expected drug concentration before and after a rapidly administered dose of drug (*Bloom et al., 1999*; *Chen et al., 1997*; *Stein, Risinger & Bloom, 1999*). Generally that approach relies on a drug with rapid onset and fading of effect, such as nicotine or cocaine. The second strategy could be called the pure pharmacodynamic method, which seeks voxels whose time-signal curve correlates with that of a clinically evident effect such as analgesia or intoxication (*Breiter et al., 1997*; *Wise et al., 2002*). The pure pharmacodynamic methods generally cannot disentangle pure pharmacologic effects of drug from the clinical effect caused by the drug. In other words, they cannot determine whether tissue in an identified voxel is showing a direct response to drug or would respond similarly to the clinical effect (e.g., pain relief) whether or not the targeted receptor was activated. Additionally, while useful for spatially mapping responses, neither of these methods allows one to compute quantitative pharmacodynamics in a single subject.

The new method presented here takes a different approach using combined pharmacokinetic-pharmacodynamic (PK-PD) modeling. This approach rests on three fundamental concepts: the nonlinearity of drug response; repeated doses of challenge drug in an interval that is brief compared to the waning of effect of a single dose of drug and to artifactual signal; and exploitation of a variable easily quantified in any imaging experiment: time.

Another substantial difference is that this method aims at deriving within-subject $EC_{50}$s. In the traditional approach to derive a population $EC_{50}$, each subject contributes to a single data point. In fact, each data point usually is derived from at least 3–5 subjects each exposed to the same dose, so the dose-response curve for the overall experiment requires numerous subjects in order to sample a wide range of doses. The population approach has occasionally been used with phMRI (*Black et al., 2010*; *Kofke et al., 2007*; *Wise et al., 2002*). The population approach is an excellent choice when the population under study is homogeneous, such as an inbred strain of mice, or when population inference is desired, such as when one is less interested in between-subject variability than in central tendency. However, the population approach requires giving some subjects doses much higher than the $EC_{50}$, which may be difficult in early human studies. The approach described here may also be a better choice when individual responses are important, because otherwise numerous imaging sessions would be required over a wide range of doses, including at doses likely to produce side effects.

## Limitations of the method

Our approach shares two limitations common to any pharmacologic activation method. First, the location of the drug effect need not occur in the physical location where the receptors are situated. A classic example is the hypothalamic-pituitary axis. The prolactin inhibiting factor, dopamine, acts at dopamine receptors on neuronal cell bodies in the hypothalamus. However, the endocrine and metabolic effects of this drug activation occur outside the brain proper, at the termination of the neuronal axonal processes in the pituitary (*Schwartz et al., 1979*). In other words, as with any pharmacologic activation approach, the method described here maps not drug receptors but rather the "downstream" effects of the drug at axon termini of activated neurons (*Ackermann et al., 1984*; *Eidelberg et al., 1997*; *McCulloch, 1982*; *McCulloch, 1984*; *Raichle, 1987*). Existing methods (such as receptor-radioligand PET) can more precisely map receptor location. However, this property of pharmacologic activation also has advantages. First, it maps "real" areas of interest (e.g., sites where drugs acting on subcortical receptors exert influence on cortical activity) (*Schwarz et al., 2004*). Second, pharmacologic activation can detect functional alterations in drug-modulated neuronal circuits even when receptor binding remains normal. Pharmacologic activation studies of dopaminergic denervation have demonstrated such effects (*McCulloch, 1982*; *McCulloch, 1984*; *McCulloch & Teasdale, 1979*; *Trugman & James, 1992*). This makes sense given that changes in second messenger function or "downstream" neurons can also modulate drug-sensitive neuronal circuits. Pharmacologic activation is the method of choice for detecting overall effects on a neuronal circuit rather than at a single level (e.g., receptors).

A second limitation of pharmacologic activation is that nonquantitative input data may affect interpretation of the results. For instance, in a blood flow PET experiment analyzed in the usual nonquantitative fashion (normalizing whole-brain mean image intensity to a constant value), the D2-like dopamine agonist pramipexole appeared to cause cerebral blood flow (CBF) increases in occipital cortex and cerebellum. However, quantitative blood flow methods revealed that these apparent increases were artifactual; pramipexole actually decreases CBF in most of the brain (preferentially in frontal cortex) while *sparing* occipital cortex and cerebellum (*Black et al., 2002*).

In addition, the novel method described herein is limited by how well its assumptions fit reality. For instance, we have modeled drug elimination but not drug distribution. After an intravenous bolus dose, many drugs show an initial rapid clearing from plasma, assumed to reflect distribution of drug into tissues, followed by the slower decline attributed to elimination (*Holford & Sheiner, 1982*). Given the short time scale on which the novel method relies, drugs with a prominent distribution phase may require the distribution half-life to be modeled separately, or instead of the elimination half-life. For instance, the elimination half-life ($t_{1/2}$) of levodopa is 1–2 h, whereas its initial distribution half-life ($t_{1/2\alpha}$) is only about 8 min (*Gancher, Nutt & Woodward, 1987*; *Nutt, Woodward & Anderson, 1985*). As a second example, many drugs show hysteresis, i.e., the effect of a drug at a given moment is influenced not only by the drug concentration at that moment but also by its concentration prior to that moment (*Contin et al., 2001*; *Contin et al., 1994*;

*Holford & Sheiner, 1982*; *Tedroff et al., 1992*). Modeling distribution half-life and hysteresis may be important for accurate parameter estimation for certain drugs.

Here we modeled four equal doses of the challenge drug. The method has obvious extensions to unequal dosing (e.g., 0.001 mg, 0.01 mg, 0.1 mg, 1 mg) or to a different number of doses. Such changes may increase the range over which the method can return accurate results, or increase the number of PK-PD parameters that can be modeled. However, in preliminary simulation work, four to eight equal doses seem to give good results. The logical extreme of more, smaller doses is a continuous slow infusion. However, if there is any delay from blood concentration to effect, including hysteresis effects, a continuous infusion may confound time and $EC_{50}$.

This method also requires rapid administration and absorption of the drug. This represents an important practical limitation of the method, as most drugs in clinical use are optimized for oral use and once- or twice-a-day dosing. The simulations and proof-of-principle biological data used brief intravenous infusions, and many drugs of interest have no marketed i.v. formulations. On the other hand, i.v. administration is not the only potential delivery option for the new method; it would also be expected to work with other rapid delivery methods such as oral or nasal inhalation or sublingual or subcutaneous administration.

## Limitations of the proof-of-principle data

The 4 monkey experiments were intended as proof-of-principle demonstrations and as such did not include the full set of controls that a more complete study would require. For instance, no vehicle infusions were included as a strong control for the drug infusions, to rule out the possibility that BOLD signal changed due to the sedated animal perceiving the investigator's activity or the infusion of fluid related to each dose. On the other hand, the signal changes in response to infusions (estimated $|E_{max}| \approx 30 = 3\%$ of modal whole-brain signal) are quite large in comparison to typical BOLD responses to most sensory or cognitive manipulations (around 0.5%). The lack of BOLD response during the first several minutes of the imaging session, before the first dose of drug was administered, may also be considered an internal control supporting the validity of the results.

QuanDyn$^{TM}$ did not identify a significant effect of drug for 6 of the 8 time-signal curves (Figs. 10A, 10B). This is of course the desired output if these volumes of interest had no meaningful response to the drug. It could also reflect excessive noise in the measurement system compared to any true signal, a problem that could be mitigated by improved imaging methods or larger *a priori* VOIs. More problematic would be a failure to identify significant drug effect because of signal artifacts not well described by quadratic drift and random error. Nevertheless, those regions in which the method identified a significant drug response showed reasonable time-signal curves (Figs. 10C, 10D) and internally consistent values for $EC_{50}$.

The midbrain response in animal 2 was positive (Fig. 10C) whereas the striatal response in the same animal was negative (Fig. 10D). The opposite sign of the responses in the two significant regions may seem surprising at first glance, since the drug presumably

has similar effects on receptors wherever they are located. However, the remote effects of the drug may depend on where excitatory or inhibitory neurotransmitters appear in the affected circuitry "downstream" to the receptor. Negative BOLD responses to a stimulus have been well studied and often reflect a decrease in blood flow and oxygen metabolism. See *Black et al. (2010, Fig. 4 and Table S7)* for a real-world phMRI example of this phenomenon with a thorough discussion.

## Applications and future directions

Simulations show that this new method can determine quantitative pharmacodynamic parameters such as $EC_{50}$ even if the underlying data derive from a nonquantitative imaging method that includes noise and slow baseline signal drift. BOLD-sensitive fMRI data appear to fit this model reasonably well. However, it may prove in practice that BOLD or other methods have additional artifactual signal changes that overcome this robustness. Other nonquantitative imaging methods may outperform BOLD-sensitive fMRI (*Chen et al., 2001*). Results may improve further using quantitative or semiquantitative methods (e.g., blood flow PET with arterial sampling, or arterial spin label perfusion MRI).

We envision several potential research applications of this method. Examples pertinent to our prior work include between-group tests of dopamine theories of drug abuse, schizophrenia, dystonia, Tourette syndrome, and complications of Parkinson disease. Application to other pharmacologic systems and other organs is equally feasible.

Several clinical applications also suggest themselves. Since the method can predict $EC_{50}$s higher than the peak blood level achieved during testing, this approach may find uses for individualized dosing estimates for drugs that are very expensive or have narrow therapeutic windows. Pure pharmacokinetic modeling has found more limited application. Another potential application would be for individualized dose-finding for drugs whose clinical response may take weeks, such as antidepressants.

Finally, there are possible applications to drug development. Assume for instance that an acute brain imaging effect occurs at the same blood level of an antidepressant that provides efficacy in chronic treatment for major depression. Then if phase I studies show a reasonable safety profile across a given range of blood concentrations, those concentrations could be used to measure $EC_{50}$ for the acute effect *in a single day* from a modest sample of patients. This would reasonably narrow the likely efficacious dose range, potentially providing substantial savings of time and money. If the receptor system being targeted shows similar sensitivity in patients and healthy controls, the initial dose-ranging estimation could even be performed in healthy subjects.

### Funding

This study was supported by the U.S. National Institutes of Health (R01 NS044598). The funders had no role in study design, data collection and analysis, decision to publish, or preparation of the manuscript.

## Grant Disclosures
The following grant information was disclosed by the authors:
U.S. National Institutes of Health: (R01 NS044598).

## Competing Interests
KJB and JMK hold a patent related to this method (U.S. Patent #8,463,552 "Novel methods for medicinal dosage determination and diagnosis"). KJB is an Academic Editor for PeerJ.

## Author Contributions
- Kevin J. Black conceived and designed the experiments, performed the experiments, analyzed the data, wrote the paper.
- Jonathan M. Koller performed the experiments, analyzed the data, contributed reagents/materials/analysis tools, reviewed and critiqued the manuscript.
- Brad D. Miller performed the experiments, analyzed the data, reviewed and critiqued the manuscript.

## Animal Ethics
The following information was supplied relating to ethical approvals (i.e., approving body and any reference numbers):

Washington University Animal Studies Committee, protocols # 20020085, 20050126.

## Patent Disclosures
The following patent dependencies were disclosed by the authors:

KJB and JMK hold a patent on this method (U.S. Patent # 8,463,552, "Novel methods for medicinal dosage determination and diagnosis").

## Data Deposition
The following information was supplied regarding the deposition of related data:

These simulated data sets (1000 instances of *voxel*($t$) for each set of parameter values and noise level) are available at datadryad.org (*Black, Koller &Miller, 2013*).

Dryad DOI: doi:10.5061/dryad.s3443.

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
