# Peer review of "Rapid quantitative pharmacodynamic imaging by a novel method: theory, simulation testing and proof of principle"

_PeerJ, doi:10.7717/peerj.117_

## Round 0.1 · original submission · Major Revisions

If you opt to submit a revised manuscript, please include formatting cues (i.e. bold or italics) to identify the specific passages of text where the reviewers' concerns are addressed. Please also consider the comments of both reviewers with respect to clarity and succinctness of text.

·

Basic reporting

The article overall might be reviewed for succinctness and clarity of exposition.

Experimental design

The simulations were extensive and addressed the major experimental variables likely to impact the utility of the method in practice.

More intuitively for many readers, the fit accuracy could also have been quantified by the number of times the fit value was within 10% (say) of the correct value, or by the average % difference between the estimated and true values. Testing with only discrete allowable parameter values, and quantifying the fitting performance in terms of sensitivity/specificity and PPV, seems a little counter-intuitive on a first read, although after some thought it does provide an interesting quantification of the frequency with which a given estimate is “good,” something the average error would not capture directly.

However, the experimental data demonstrating the utility of the method in practice is limited.

Validity of the findings

Limited by little actual experimental data included, and for much of that the method did not seem to work very well, seemingly limited by the SNR obtained in practice.

No control condition (e.g., vehicle injections) were included. What do the BOLD time courses look like during sequential vehicle injections? Such a control condition would be helpful to assess the likely incidence of false positive fits in practice, and the time courses might also be informative to the confounds (null model) included in the simulations.

Additional comments

This is an innovative piece of work that addresses an important gap in pharmacological imaging, namely the exploration of dose-response relationships and the generation of robust brain-region specific pharmacological parameters, such as EC50, from the imaging data. The approach represents an admirable example of creative thinking about this problem, and actually opens up a wide range of interesting variants to explore both experimentally and mathematically, as the authors allude to in the Discussion. The proposed method is a very nice idea that deserves dissemination.

However, I suggest the authors address the following points more explicitly in the paper:

- The method, involving rapid sequential injections of a compound, appears intrinsically limited to compounds with intravenous routes of administration. This may not be unduly limiting in preclinical experiments but would be a limitation in human studies, where most compounds are formulated for oral administration and designed for pharmacokinetics suitable for convenient timing (e.g. daily) of dosing. The authors should qualify their final paragraph to this effect.

- The method generates within-subject EC50 estimates, an approach that is probably optimal for monkey studies where the number of subjects used is very small, and the information extracted from each individual must be maximized. This can be contrasted with more standard approaches to estimating pharmacodynamic parameters, in which each subject is a single data point but the overall experiment samples a wide range of doses (and hence exposure to compound), enabling dose-response and exposure-response (e.g., by using sigmoid models) relationships to be determined. This is widely used both preclinically and clinically, but has found very limited application to pharmacological imaging to date. Although less innovative per se, this might be a more generally tractable approach to quantitative pharmacology from imaging data in rodents and humans. For context, the authors could compare and contrast these approaches, their relative strengths and weaknesses, and when one might choose one over the other.

- Despite its promise, based on the limited experimental data included in the manuscript the method did not appear to work robustly in practice. This may represent the “real world” but it would be most helpful for readers of the paper if statistics for the full set of time courses were tabulated and shown graphically (not just the ones with the good fits).

- The authors should also discuss the positive vs. negative signal responses observed in the different brain regions (Fig. 10).

- Based on the seeming difficulty of the method to work reliably in practice, the authors could suggest some ways to improve the applicability of the method. (E.g., might it be more applicable to compounds with different mechanisms allowing larger VOIs? What can we learn from the in vivo data regarding the assumptions made in the modeling, especially regarding confound signals – were these adequately modeled in the simulations?) If the authors have any additional in vivo data that more convincingly demonstrate the applicability of the method in practice, they would be a welcome addition to the paper.

- The single dose experiment described in the Methods is not reported on in the Results. This discrepancy should be resolved.



Minor points for clarification:

Methods, p.6, l.37: “1.0-2.0% isoflurane” – this seems quite a wide range, please explain. Is the level titrated individually? What is the MAC in the macaque?

Methods, p.7, l.26: “infused rapidly” – please specify the duration of each infusion.

Figures: Several of the figures seem to be missing axis annotations on the graphs.

·

Basic reporting

1. Quality of the most figures needs to be improved.
2. Number of figures seems more than usual, some figures could be consolidated or removed.

Experimental design

The information is sufficient. However, it appears longer than usual.

Validity of the findings

This is carefully designed experiment and tried to address an important issue related to phMRI. No doubt, the experiment was conducted based on a novel idea and executed relatively well.

Overall, the data is solid and sounds although in vivo data looks a little bit weak. The discussion was well written with Pro and Con about the novel methods.

Additional comments

Authors conducted a nice but complex fMRI study based on a novel idea and tried to address an important issue associated phMRI, in which every phMRI investigators are interested in. The review would like to have better figures with the manuscript and to have a simplified methods which was a little bit hard to follow.

Overall, it is well written manuscript with very interesting idea and resutl.

---

## Round 0.2 · accepted · Accept

The responses to reviewers' comments were thorough, and I believe the manuscript is considerably improved based upon that interaction. I look forward to seeing the published article, and I hope you will continue to submit your work to PeerJ.

Reviewer 1 ·

Basic reporting

No new comments for the revised manuscript.

Experimental design

No new comments for the revised manuscript.

Validity of the findings

No new comments for the revised manuscript.

Additional comments

After carefully reading the revised manuscript, tables and figures, the reviewer believes that the authors has properly addressed all major concerns or questions raised by the reviewer, and made properly changes on the tables and figures, and made good arguments on those issue which were hard to be changes. In addition, after considering the challenges and difficulties that the authors facing regarding the complexity of the study, the reviewer agrees to accept the manuscript for the publication.

·

Basic reporting

The authors have improved the clarity of the article in their revisions.

Experimental design

The design is adequate for the proof-of-principle nature of this experiment. I look forward to seeing the results of more extensive experimental studies in future publications.

Validity of the findings

Limited by the proof-of-principle nature of this paper, but sufficient to encourage - I hope - interest in this novel approach.

Additional comments

The authors have thoroughly addressed the concerns raised in the reviews.